# Becker muscular dystrophy mice showed site-specific decay of type IIa fibers with capillary change in skeletal muscle

**Daigo Miyazaki[1,2]\*[†], Mitsuto Sato[1,3], Naoko Shiba[4], Takahiro Yoshizawa[5], Akinori Nakamura[6,7]\***

[1]Department of Medicine (Neurology and Rheumatology), Shinshu University School of Medicine, Matsumoto, Japan; [2]Intractable Disease Care Center, Shinshu University Hospital, Matsumoto, Japan; [3]Department of Brain Disease Research, Shinshu University School of Medicine, Matsumoto, Japan; [4]Department of Regenerative Science and Medicine, Shinshu University, Matsumoto, Japan; [5]Research Center for Advanced Science and Technology, Shinshu University, Matsumoto, Japan; [6]Department of Clinical Research, NHO Matsumoto Medical Center, Matsumoto, Japan; [7]Third Department of Medicine, Shinshu University School of Medicine, Matsumoto, Japan

**\*For correspondence:**
miyajiro@shinshu-u.ac.jp (DM);
anakamu@shinshu-u.ac.jp (AN)

**Present address:** [†]Department of Medicine (Neurology and Rheumatology), Shinshu University School of Medicine, Matsumoto, Japan

**Competing interest:** The authors declare that no competing interests exist.

## eLife Assessment

The authors present three transgenic models carrying three representative exon deletions of the dystrophin gene. The findings presented are **valuable** to the field of muscle diseases, particularly muscular dystrophies. The evidence provided in the manuscript is **convincing**, with rigorous biochemical assays and state-of-the-art microscopy methods.

**Abstract** Becker muscular dystrophy (BMD), an X-linked muscular dystrophy, is mostly caused by an in-frame deletion of Duchenne muscular dystrophy (DMD). BMD severity varies from asymptomatic to severe, associated with the genotype of DMD. However, the underlying mechanisms remain unclear. We established BMD mice carrying three representative exon deletions: ex45–48 del., ex45–47 del., and ex45–49 del. (d45–48, d45–47, and d45–49), with high frequencies and different severities in the human BMD hotspot. All three BMD mice showed muscle weakness, muscle degeneration, and fibrosis, but these changes appeared at different times for each exon deletion, consistent with the severities obtained by the natural history study of BMD. BMD mice showed site-specific muscle changes, unlike *mdx* mice, which showed diffuse muscle changes, and we demonstrated selective type IIa fiber reduction in BMD mice. Furthermore, BMD mice showed sarcolemmal neuronal nitric oxide synthase (nNOS) reduction and morphological capillary changes around type IIa fibers. These results suggest that capillary changes caused by nNOS reduction may be associated with the mechanism of skeletal muscle degeneration and type IIa fiber reduction in BMD mice. BMD mice may be useful in elucidating the pathomechanisms and developing vascular targeted therapies for human BMD.

## Introduction

Becker muscular dystrophy (BMD), an X-linked muscle disorder, is characterized by progressive muscle wasting and weakness, mostly caused by an in-frame variant in *DMD* encoding the sarcolemmal

protein dystrophin. In BMD muscle tissues, truncated and reduced dystrophin is expressed; therefore, its clinical status is generally milder than that of Duchenne muscular dystrophy (DMD), an allelic disorder with complete loss of dystrophin (*Koenig et al., 1989*). BMD is clinically heterogeneous, with some affected individuals experiencing a near-normal lifestyle and lifespan, while others lose the ability to walk in their late teens or early 20s (*Beggs et al., 1991*; *Bushby and Gardner-Medwin, 1993*; *Comi et al., 1994*).

A natural history study of patients with BMD revealed that part of the skeletal muscle phenotype may be associated with the genotype of *DMD* and that there are several patterns of exon deletions associated with severe or milder phenotypes (*Nakamura et al., 2023*). In-frame exon deletions of BMD accumulate in exons 45–55 (first hotspot) or exons 3–7 (second hotspot) of *DMD*. Particularly, 80% of all in-frame deletions were included in the first hotspot: exons 45–55. Among these, the deletion of exons 45–47 (d45–47) is the most frequent, approximately 30% of patients with BMD have this in-frame deletion. Deletion of exons 45–48 (d45–48) is the second most frequent; approximately 18% of patients with BMD have this deletion and a milder phenotype than D45–47. In contrast, the deletion of exons 45–49 (d45–49) has a more severe phenotype than D45–47 (*Nakamura et al., 2023*).

The relationship between the pathomechanisms of disease severity and each exon deletion remains unclear. In the muscle tissue of patients with BMD, truncated dystrophin expression levels are lower than those of healthy controls, and dystrophin immunohistochemistry shows a 'faint and patchy' staining pattern (*Beggs et al., 1991*; *Jimi et al., 1992*). The correlation between the severity and expression levels of truncated dystrophin is still conflicting, and the truncated dystrophin expression levels in muscle tissues are well correlated with the clinical severity of BMD (*Anthony et al., 2011*). However, there is another report that truncated dystrophin levels do not appear to be a major determinant of disease severity in BMD (*van den Bergen et al., 2014*).

Dystrophin is a large filamentous protein with a molecular weight of 427 kDa that protects the sarcolemma from mechanical stress during muscle contraction. Dystrophin connects to cytoskeletal protein families that can assemble into macromolecular structures with a large number of proteins and lipids (*Le Rumeur et al., 2010*). Recently, a report suggested that the BMD clinical heterogeneity is associated with the changes in dystrophin structure generated by exon deletion using an in silico prediction model (*Nicolas et al., 2015*). They expect that the d45–48 and d45–51 displayed a structure similar to that of wild-type (WT) dystrophin (hybrid repeat), whereas the d45–47 and d45–49 lead to proteins with an unrelated structure (fractional repeat) among the representative exon deletions in the BMD variant hotspot. Exon deletions associated with fractional repeats are expected to lead to a more severe BMD phenotype than those associated with hybrid repeats; however, the influence of changes in dystrophin structure on clinical BMD severity is not fully understood.

Recently, two animal models of BMD have been developed. One group generated the first BMD rat model carrying a deletion of exons 3–16 of the rat *Dmd* using CRISPR/Cas9 (*Teramoto et al., 2020*). This BMD rat model exhibited muscle degeneration, muscle fibrosis, heart fibrosis, and reduced truncated dystrophin levels. Another group generated a BMD mouse model carrying a deletion in exons 45–47 of the mouse *Dmd* (*Heier et al., 2023*). The BMD mice show skeletal muscle weakness, heart dysfunction, increased myofiber size variability, increased centronuclear fibers, and reduced truncated dystrophin. However, underlying mechanism of BMD severity remains unclear, and comparative investigations using several BMD animal models carrying exon deletions associated with various severities are necessary.

To clarify the issue, we generated three types of BMD mice with representative exon deletions in the BMD hotspots: d45–47, d45–48, and d45–49 using CRISPR–Cas9 genome editing and compared the phenotypes and histopathological changes.

## Results

### Establishment of BMD mice carrying d45–48, d45–47, and d45–49 in *Dmd*

To create mouse strains carrying three in-frame deletions: d45–48, d45–47, and d45–49, we used CRISPR/Cas9 genome editing technique to introduce approximately 160,000 bp genomic deletions into the endogenous murine *Dmd*. Guide RNAs (gRNAs) were designed to target protospacer adjacent motif sequences upstream of *Dmd* exon 45 and downstream of *Dmd* exons 47, 48, and 49. We

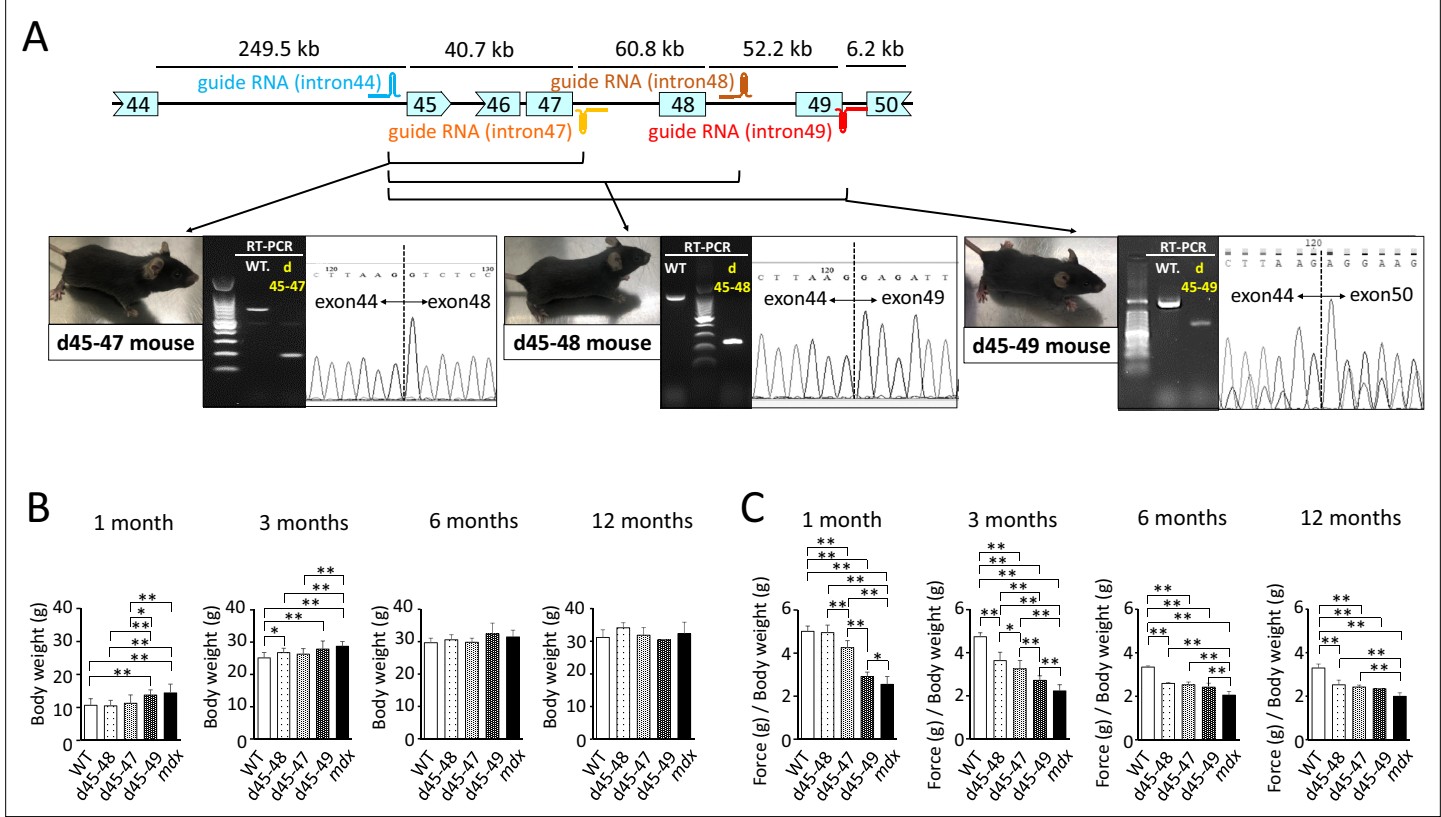

**Figure 1.** Generation and phenotypic changes in Becker muscular dystrophy (BMD) mice carrying d45–47, d45–48, and d45–49 in-frame mutation in mouse *Dmd*. (**A**) Schematic representation of mouse *Dmd* from exons 44 to 50, and the location of guide RNAs. We established d45–47, d45–48, and d45–49 BMD mouse models, using guide RNA corresponding to the sequences of introns 44 and 47, introns 44 and 48, and introns 44 and 49, respectively. All BMD mouse models were confirmed having desired mutations by RT-PCR. (**B**) Body weight (g) in wild-type (WT), d45–48, d45–47, d45–49, and *mdx* mice, at the ages of 1, 3, 6, and 12 months (*n* = 10 at 1 and 3 months, *n* = 4 at 6 and 12 months). (**C**) Relative (to body weight (g)) forelimb grip strength in WT, d45–48, d45–47, d45–49, and *mdx* mice, at the ages of 1, 3, 6, and 12 months (*n* = 10 at 1 and 3 months, *n* = 4 at 6 and 12 months). Bar: mean ± SD; *p < 0.05, **p < 0.01.

The online version of this article includes the following video and figure supplement(s) for figure 1:

**Figure supplement 1.** Intronic deletion breakpoint in mouse *Dmd*, and the result of the hanging wire test of Becker muscular dystrophy (BMD) mice carrying d45-47, d45-48 and d45-49.

**Figure 1—video 1.** The movie of the results of hanging wire test in wild-type (WT), d45–48, d45–47, d45–49, and *mdx* mice, at the age of 3 months.
https://elifesciences.org/articles/100665/figures#fig1video1

confirmed the genomic deletion of dystrophin exons 45–47, 45–48, and 45–49 by cDNA sequencing (*Figure 1A*). By analyzing the deletion breakpoint, we confirmed that double-strand breaks were formed inside the sequences corresponding to gRNAs in d45–48, d45–47, and d45–49 mice (*Figure 1—figure supplement 1A*). We performed phenotypic testing of WT, d45–48, d45–47, d45–49, and *mdx* mice at 1, 3, 6, and 12 months of age. d45–49 and *mdx* mice had large body weights at 1 and 3 months of age, but the d45–48 and d45–47 mice did not differ from that in WT mice (*Figure 1B*).

## d45–49 mice showed muscle weakness at an earlier age compared to d45–48 and d45–47 mice

To assess the muscle strength of the BMD mice, we conducted forelimb grip tests at 1, 3, 6, and 12 months of age, and the hanging wire tests at 3 months of age, and compared with WT and *mdx* mice. At the age of 1 month, d45–47, d45–49, and *mdx* mice showed reduced forelimb grip strength compared to WT and d45–48 mice. However, the weakness of d45–47 mice was milder compared to d45–49, and *mdx* mice of the same age were used. At the age of 3 months, reduced grip strength in the d45–47 mice was remarkable, and d45–48 mice exhibited a slight reduction in forelimb grip

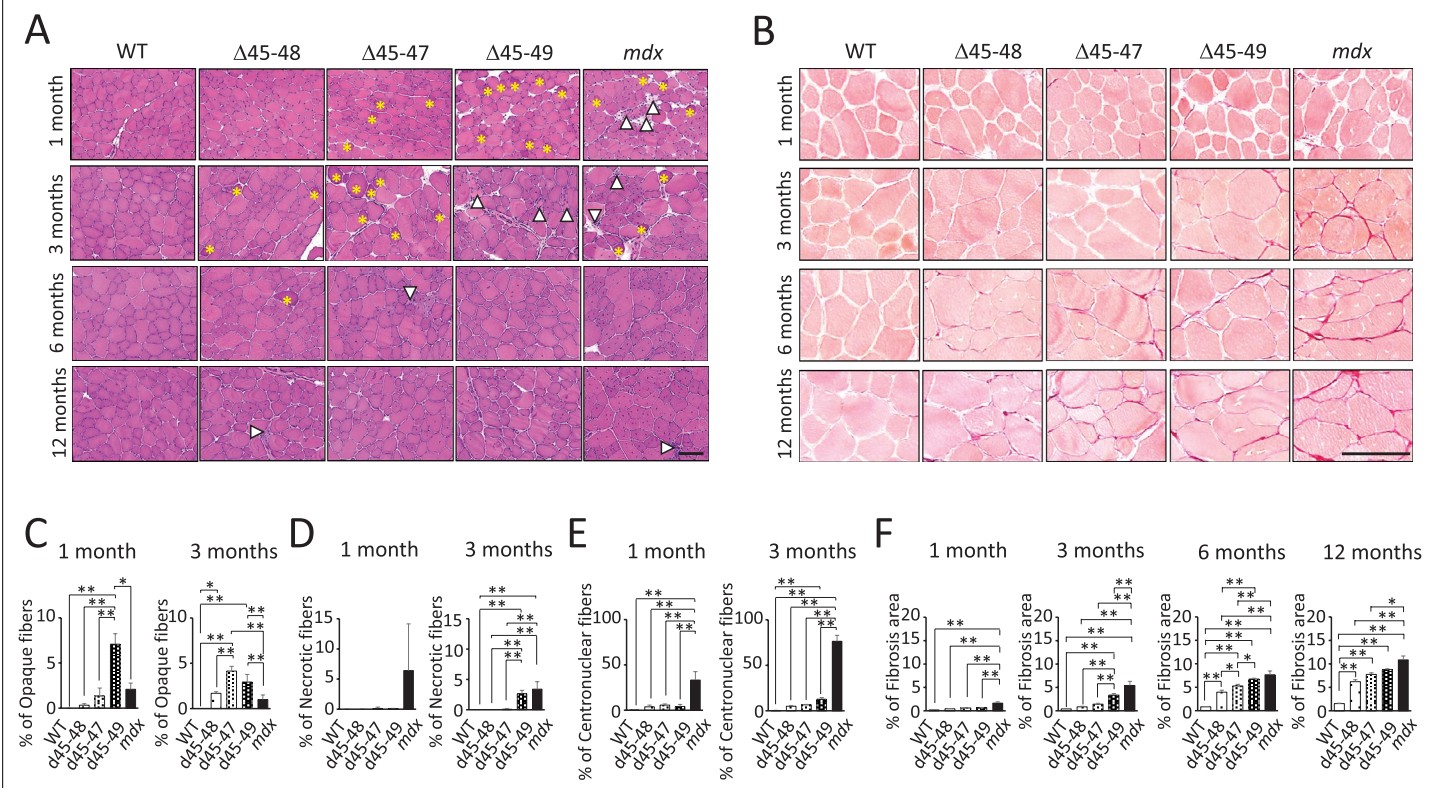

**Figure 2.** Histopathological changes in Becker muscular dystrophy (BMD) mice carrying d45–47, d45–48, and d45–49. (**A**) Hematoxylin–eosin stain of tibialis anterior (TA) muscles in wild-type (WT), d45–48, d45–47, d45–49, and *mdx* mice, at the ages of 1, 3, 6, and 12 months. Yellow asterisk: mean opaque-fibers. Arrow head: mean necrotic fibers. (**B**) Sirius Red stain of TA in WT, d45–48, d45–47, d45–49, and *mdx* mice, at the ages of 1, 3, 6, and 12 months. (**C**) Percent opaque fibers at TA in WT, d45–48, d45–47, d45–49, and *mdx* mice, at the ages of 1 and 3 months (*n* = 3). (**D**) Percent necrotic fibers at TA in WT, d45–48, d45–47, d45–49, and *mdx* mice, at the ages of 1 and 3 months (*n* = 3). (**E**) Percent centronuclear fibers at TA in WT, d45–48, d45–47, d45–49, and *mdx* mice, at the ages of 1 and 3 months (*n* = 3). (**F**) Percent fibrosis area at TA in WT, d45–48, d45–47, d45–49, and *mdx* mice, at the ages of 1, 3, 6, and 12 months (*n* = 3). Bar: mean ± SD; *p < 0.05, **p < 0.01. Scale bar means 100 µm.

The online version of this article includes the following figure supplement(s) for figure 2:

**Figure supplement 1.** Histopathological changes of diaphragm and cardiac muscles, and the time-coursed histopathological changes of the skeletal muscle in Becker muscular dystrophy (BMD) mice carrying d45-47, d45-48 and d45-49.

strength. Finally, after 6 months, all three BMD mice reached the same levels of reduced forelimb grip strength, although not as much as that of *mdx* mice (*Figure 1C*). The hanging wire test showed reduced suspension times in the d45–49 and *mdx* mice, although the WT, d45–48, and d45–47 mice exceeded the 600 s time limit (*Figure 1—figure supplement 1B*). Furthermore, the observation using video for the hanging wire test revealed changes in the mice moving during wire grasping: WT, d45–48, and d45–47 mice could quickly bring their posterior legs up toward their faces and catch the wire; in contrast, d45–49 and *mdx* mice could not raise their posterior legs above their chest (*Figure 1—video 1*).

## d45–49 mice showed earlier muscle degeneration and fibrosis compared with d45 –48 and d45–47 mice

Hematoxylin–eosin (HE) staining of the tibialis anterior (TA) muscles showed muscle degeneration and regeneration (i.e., opaque fibers, necrotic fibers, and centronuclear fibers) in all BMD and *mdx* mice, except for WT mice (*Figure 2A*). Opaque fibers, which are thought to be precursors of necrotic fibers, increased at an earlier age of 1 month in d45–49 mice compared with WT mice; in contrast, the proportion of opaque fibers differs significantly between d45–47 and WT mice at 3 months, with an increased tendency only in 1-month-old mice (*Figure 2C*). Necrotic fibers were seen only in *mdx* mice at 1 month, but were also increased in d45–49 mice at 3 months (*Figure 2D*). Centronuclear fibers,

which are thought to be regenerated fibers after muscle damage, were increased in *mdx* mice and slightly increased in all BMD mice at 1 month. At 3 months, centronuclear fibers were increased in the d45–49 mice (*Figure 2E*). In addition, Sirius Red staining of the TA muscles revealed an increase in fibrosis in all BMD and *mdx* mice, except in WT mice (*Figure 2B*). At 1 month, an increased fibrotic area was seen only in *mdx* mice, but after 3 months, it was seen also in d45–49 mice. After 6 months, fibrosis gradually became apparent, even in d45–48 and d45–47 mice (*Figure 2F*).

HE and Sirius Red staining revealed increased centronuclear fibers and fibrosis in the diaphragm muscles, and increased myocardial fibrosis in the cardiac muscles in all BMD mice at 12 months (*Figure 2—figure supplement 1A, B*). However, these changes were minor compared with those observed in *mdx* mice.

All BMD mice exhibited muscle degeneration and fibrosis; however, these findings became more apparent with exon deletions: the time-course histopathological observations revealed that muscle degeneration and fibrosis deteriorated in the following order: d45–49, d45–47, and d45–48 mice (*Figure 2—figure supplement 1C*).

## All BMD mice showed a decrease in neuronal nitric oxide synthase expression and no utrophin overexpression

We confirmed the immunohistochemical analysis in TA muscles of dystrophin and dystrophin–glyco-protein complex (DGC) proteins: alpha-sarcoglycan (aSG) and neuronal nitric oxide synthase (nNOS) at 1 and 3 months. Dystrophin immunohistochemistry demonstrated a reduction of dystrophin and 'faint' and 'patchy' staining patterns in all BMD mice. Consistent with the lack of an nNOS-binding site, encoded by exons 42–45, all BMD mice showed decreased levels of nNOS expression in the sarcolemma to levels similar to *mdx* mice. On the contrary, consistent with the remaining dystroglycan-binding site: encoded by exons 63–70, the expression of aSG was detected on the sarcolemma in all BMD mice, whereas the expression levels were slightly low compared with WT mice (*Figure 3A*). Western blot (WB) analysis demonstrated truncated dystrophin expression in all BMD mice at 1 and 3 months (*Figure 3B*). The truncated dystrophin expression levels were reduced to 30–40% at 1 month, and 10–20% at 3 months compared to those in WT mice (*Figure 3E*), despite adequate levels of dystrophin mRNA expression (*Figure 3—figure supplement 1A*). By WB analysis, the utrophin expression levels showed only an increased tendency in all BMD mice at 3 months, whereas there was a significant increase in *mdx* mice (8-fold at 1 month and 30-fold at 3 months) compared to WT mice (*Figure 3C, F*). In contrast, utrophin mRNA expression levels did not differ among WT, BMD, and *mdx* mice (*Figure 3—figure supplement 1B*). Further, WB analysis demonstrated decreased levels of nNOS in all BMD mice as well as *mdx* mice at 3 months (*Figure 3D, G*), despite adequate levels of nNOS mRNA expression (*Figure 3—figure supplement 1C*).

## BMD mice showed site-specific muscle degeneration and type IIa fiber reduction

Generally, patients with muscle diseases, including BMD, show changes in MRI or CT images of a particular group of muscles (*Díaz-Manera et al., 2015*; *Tasca et al., 2012*). This allows investigators to use muscle images as a diagnostic tool for diagnosis. Thus, we examined whether the histopathological changes are different in particular parts of the muscle in BMD mice at 3 months, by observing the inner or outer part of the TA, four parts of the quadriceps: vastus lateralis (VL), rectus femoris (RF), vastus medialis (VM), vastus intermedius (VI), and three parts of the erector spinae: multifidus (MF), longissimus (LG), and iliocostalis (IC). *mdx* mice showed diffuse muscle degeneration on HE staining and no remarkable histopathological differences between the inner and outer part of the TA. In contrast, BMD mice showed significant site-specific muscle degeneration in the inner part of the TA compared with the outer part (*Figure 4A*, *upper panel*). Even in the quadriceps and erector spinae, d45–49 mice showed site-specific muscle degeneration predominantly in the RF, VM, VI, MF, and LG, whereas less muscle degeneration was observed in the VL and IC. In contrast, *mdx* mice exhibited diffuse muscle generation in all parts of the quadriceps and erector spinae (*Figure 4D, G*). Serum creatine kinase (CK) levels at 1, 3, 6, and 12 months were high in *mdx* mice, and were two- to fourfold higher in all BMD mice after 3 months compared to WT mice, although the difference was not statistically significant (*Figure 4—figure supplement 1A*). We observed site-specific muscle degeneration in BMD mice, especially in the deep muscle or inner part of the same muscle. Next,

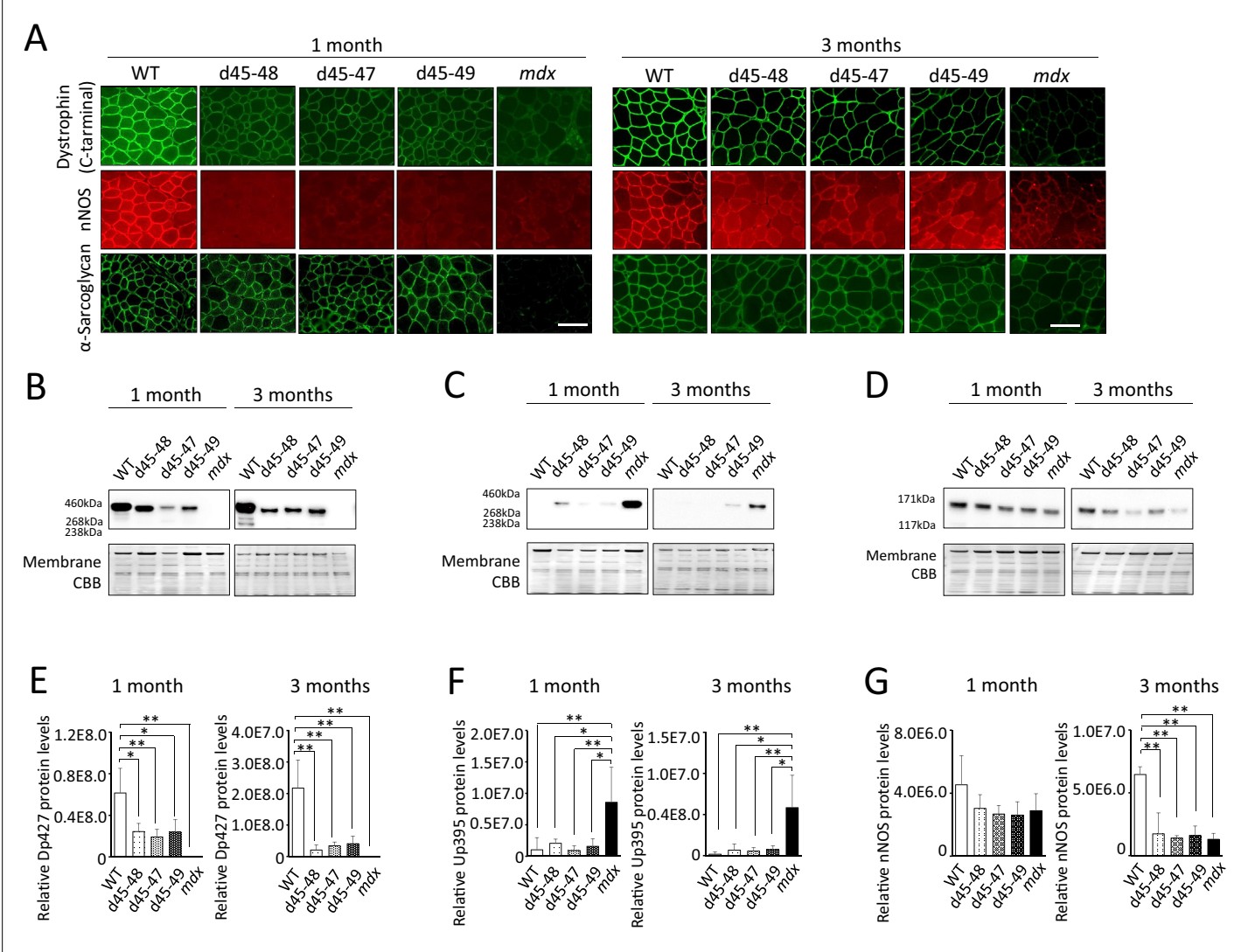

**Figure 3.** All Becker muscular dystrophy (BMD) mice showed truncated dystrophin expression and decreased sarcolemmal neuronal nitric oxide synthetase (nNOS) expression. (**A**) Dystrophin, nNOS, and alpha-sarcoglycan immunohistochemistry of tibialis anterior (TA) in wild-type (WT), d45–48, d45–47, d45–49, and *mdx* mice, at the age of 1 and 3 months. (**B**) Western blot analysis for dystrophin in WT, d45–48, d45–47, d45–49, and *mdx* mice, at the ages of 1 and 3 months. (**C**) Western blot analysis for dystrophin-homolog utrophin in WT, d45–48, d45–47, d45–49, and *mdx* mice, at the ages of 1 and 3 months. (**D**) Western blot analysis for nNOS in WT, d45–48, d45–47, d45–49, and *mdx* mice, at the ages of 1 and 3 months. (**E**) Relative Dp427 protein levels (normalized by total protein band intensity) in WT, d45–48, d45–47, d45–49, and *mdx* mice (*n* = 3). (**F**) Relative Up395 protein levels (normalized by total protein band intensity) in WT, d45–48, d45–47, d45–49, and *mdx* mice (*n* = 3). (**G**) Relative nNOS protein levels (normalized by total protein band intensity) in WT, d45–48, d45–47, d45–49, and *mdx* mice (*n* = 3). Bar: mean ± SD; *$p < 0.05$, **$p < 0.01$. Scale bar means 100 µm.

The online version of this article includes the following figure supplement(s) for figure 3:

**Figure supplement 1.** mRNA levels of dystrophin, utrophin and neuronal nitric oxide synthetase (nNOS) in Becker muscular dystrophy (BMD) mice.

we examined the muscle fiber composition in each muscle or part of the same muscle in WT, BMD, and *mdx* mice using MYH-1, -2, -4, and -7 immunohistochemistry, corresponding to type IIx, IIa, IIb, and I fibers, respectively. In WT mice at 3 months, type IIb fibers were dominant in the outer part of the TA, while type IIa and IIx fibers were dominant in the inner part of the TA, although type I fibers were rare in the TA (*Figure 4A*, *middle and lower panels*). These deviations in muscle fiber type in the TA were also seen in BMD and *mdx* mice at 3 months, however, the ratio of type IIa fibers in the inner part of the TA was decreased in all BMD and *mdx* mice compared with WT mice. In contrast, the ratio of type IIb to type IIx fibers was unchanged in all BMD and *mdx* mice compared with that in WT mice (*Figure 4B, C*). The deviation in muscle fiber type was seen even in the quadriceps

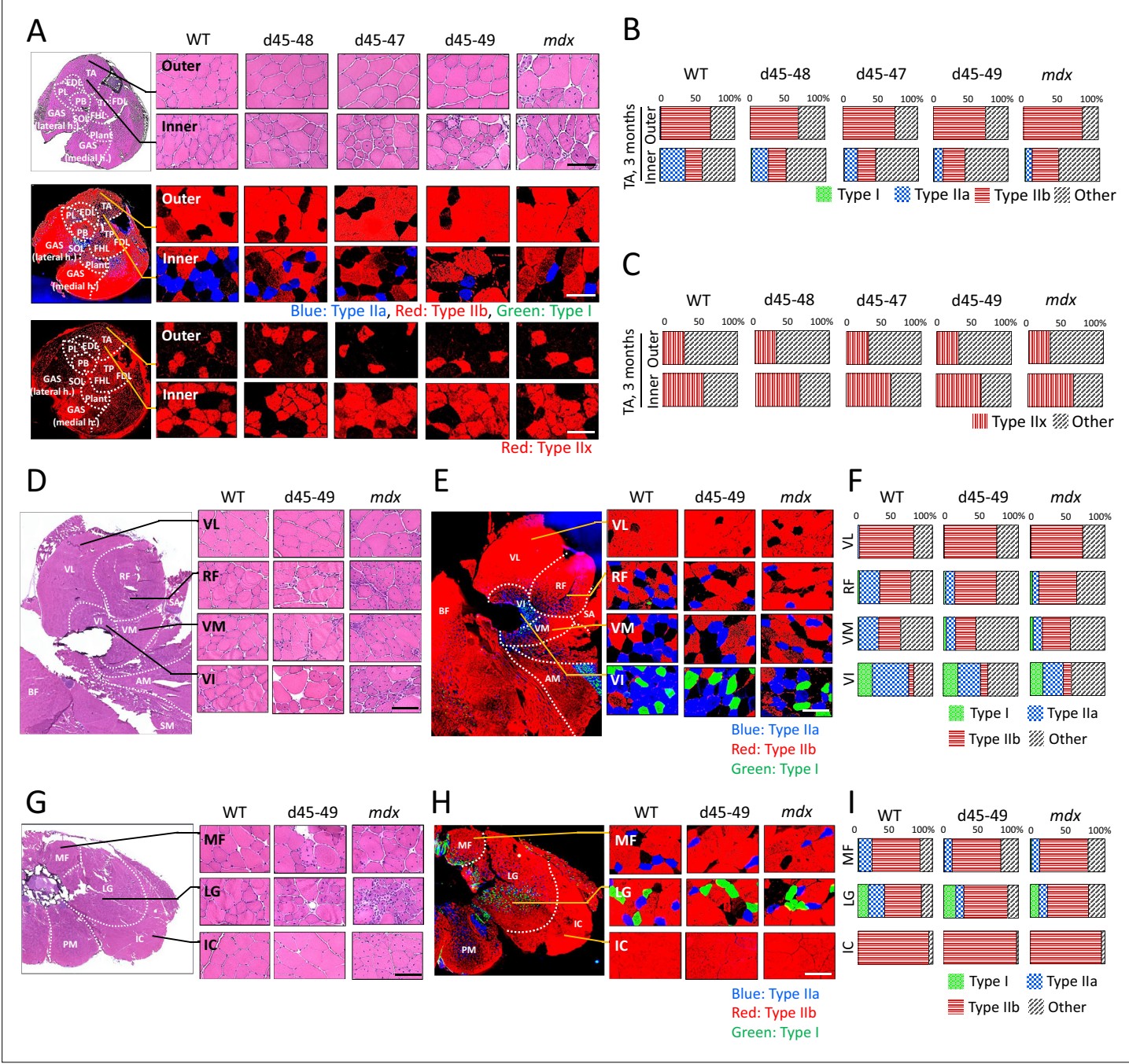

**Figure 4.** All Becker muscular dystrophy (BMD) mice showed site-specific muscle degeneration especially in inner part of tibialis anterior (TA) with type IIa fiber reduction. (**A**) Hematoxylin–eosin stain (upper panels), MYH-2, -4, and -7 immunohistochemistry (middle panels: type IIa, IIb, and I fibers, respectively) and MYH-1 immunohistochemistry (lower panels: type IIx fibers) at outer or inner part of TA in wild-type (WT), d45–48, d45–47, d45–49, and *mdx* mice, at the age of 3 months. (**B**) The proportion of muscle fiber types at outer or inner part of TA about type I, IIa, IIb, and others, in WT, d45–48, d45–47, d45–49, and *mdx* mice, at the age of 3 months (*n* = 3). (**C**) The proportion of muscle fiber types at outer or inner part of TA about type IIx and others, in WT, d45–48, d45–47, d45–49, and *mdx* mice, at the age of 3 months (*n* = 3). (**D**) Hematoxylin–eosin stain at four component muscles of quadriceps muscle: vastus lateralis (VL), rectus femoris (RF), vastus medialis (VM), and vastus intermedius (VI), in WT, d45–49, and *mdx* mice, at the age of 3 months. (**E**) MYH-2, -4, and -7 immunohistochemistry (type IIa, IIb, and I fibers [blue, red, and green, respectively]) at VL, RF, VM, and VI in WT, d45–49, and *mdx* mice, at the age of 3 months. (**F**) The proportion of muscle fiber types at VL, RF, VM, and VI in WT, d45–49, and *mdx* mice, at the age of 3 months (average of *n* = 3). (**G**) Hematoxylin–eosin stain at three component muscles of erector spinae muscle: multifidus (MF), longissimus (LG), and iliocostalis (IC), in WT, d45–49, and *mdx*, at the age of 3 months. (**H**) MYH-2, -4, and -7 immunohistochemistry immunohistochemistry (type IIa, IIb, and

*Figure 4 continued on next page*

*Figure 4 continued*

I fibers [blue, red, and green, respectively]) at MF, LG, and IC in WT, d45–49, and *mdx* mice, at the age of 3 months. (**I**) The proportion of muscle fiber types at MF, LG, and IC in WT, d45–49, and *mdx* mice, at the age of 3 months (average of *n* = 3). Scale bar means 100 µm.

The online version of this article includes the following figure supplement(s) for figure 4:

**Figure supplement 1.** Serum CK levels, cross-sectional area (CSA) of TA muscles and mRNA levels of muscle atrophy-related genes in Becker muscular dystrophy (BMD) mice.

and erector spinae in WT mice at 3 months: type IIb fibers were present dominantly in the VL and outer part of the RF and IC, and type IIa fibers were present dominantly in the inner part of the RF, VM, VI, MF, and LG. Type I fibers, which are rare in the TA, were present in the VI, MF, and LG. In particular, the VL and IC were mostly composed of type IIb fibers and rarely contained other fiber types (*Figure 4E, H*). In contrast, all BMD and *mdx* mice showed the same fiber type deviation in the quadriceps and erector spinae at 3 months, however, the ratio of type IIa fibers in the inner part of the RF, VM, VI, MF, and LG was lower than that in WT mice. In contrast, the ratio of type I and IIb fibers was unchanged or slightly increased in all BMD and *mdx* mice compared to that in WT mice (*Figure 4F, I*). We examined the cross-sectional areas (CSAs) of TA muscles in BMD and *mdx* mice to evaluate the association between type IIa fiber reduction and muscle atrophy. Their CSAs were rather high at 3 months, in accordance with muscle hypertrophy, compared with those of WT mice

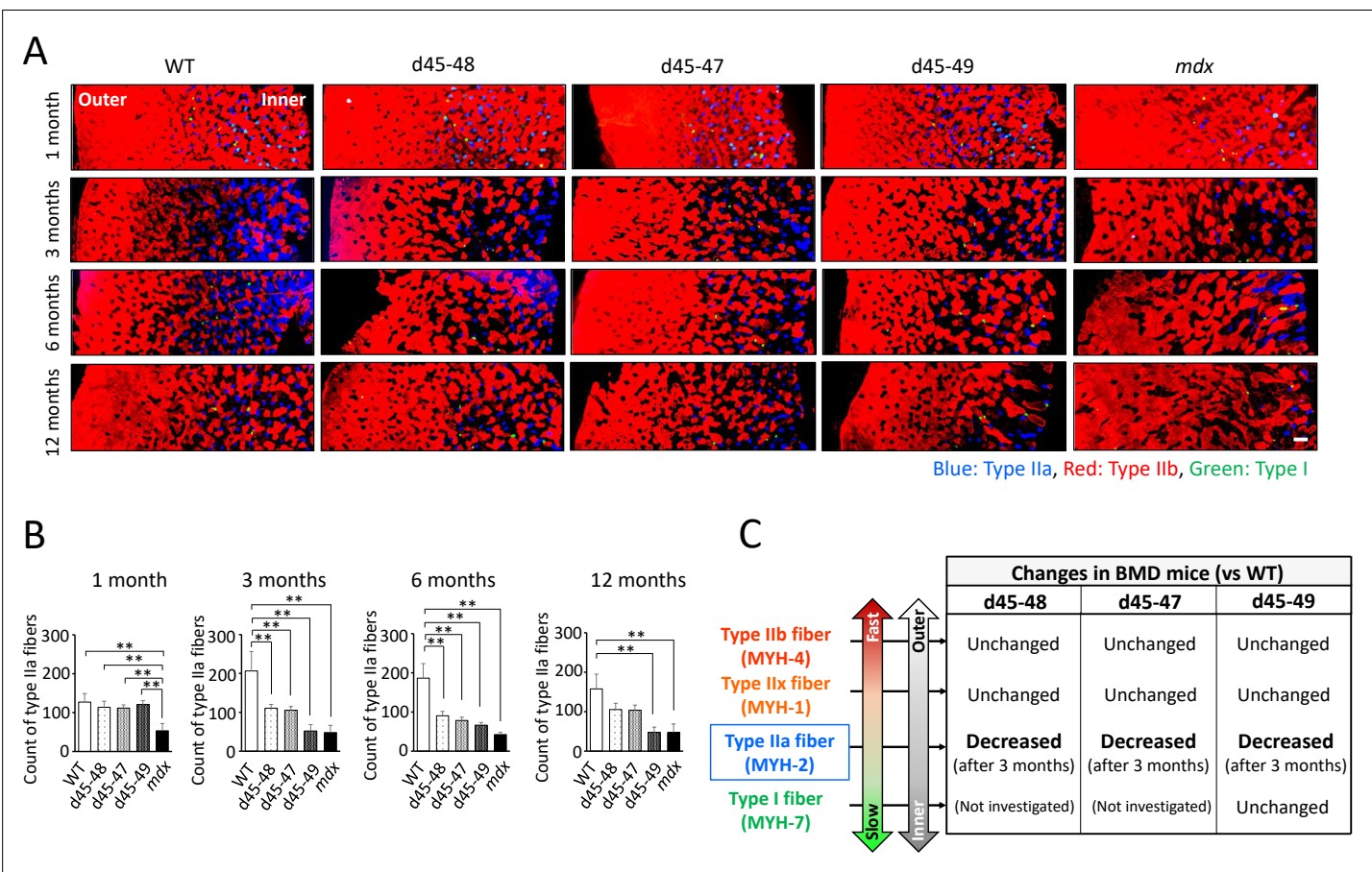

**Figure 5.** Type IIa fiber decrement started from the age of 3 months in Becker muscular dystrophy (BMD) mice. (**A**) MYH-2, -4, and -7 immunohistochemistry (type IIa, IIb, and I fibers, respectively) at tibialis anterior (TA) under low magnification in wild-type (WT), d45–48, d45–47, d45–49, and *mdx* mice, at the age of 1, 3, 6, and 12 months. (**B**) The count of type IIa fibers involved in one TA section (at the middle part of TA) in WT, d45–48, d45–47, d45–49, and *mdx* mice, at the ages of 1, 3, 6, and 12 months (*n* = 3). (**C**) Schematic representation about the character of type I, IIa, IIx, and IIb fibers and its proportional changes in BMD mice compared with WT mice. Bar: mean ± SD; **p < 0.01. Scale bar means 100 µm.

The online version of this article includes the following figure supplement(s) for figure 5:

**Figure supplement 1.** A shift of the muscle fiber type composition in a cardiotoxin (CTX) induced muscle regeneration.

(*Figure 4—figure supplement 1B*). In addition, Murf1 and atrogin-1 mRNA expression levels (representative muscle atrophy inducing factors) did not differ among WT, BMD, and *mdx* mice (*Figure 4—figure supplement 1C, D*).

## Type IIa fibers were decreased after 3 months in BMD mice

The composition of muscle fiber types changes according to postnatal development in mice (*Miwa et al., 2009*). We examined the changes in muscle fiber type composition during development in WT, BMD, and *mdx* mice, especially focusing on the type IIa fiber transition. Immunohistochemistry of MYH-1, -2, and -4 in the TA at 1, 3, 6, and 12 months revealed time-course changes in the number of type IIa fibers in WT, BMD, and *mdx* mice (*Figure 5A*). In WT mice, the count of type IIa fibers was approximately 100 fibers in a TA section at 1 month, and were twofold increased at 3 and 6 months, then was slightly down-regulated at 12 months. In contrast, the count of type IIa fibers in BMD mice was the same as that in WT mice at 1 month but decreased after 3 months, which is opposite to that of WT mice. Furthermore, in *mdx* mice, the number of type IIa fibers was low at 1 month and did not recover throughout the observation period (*Figure 5B*).

## Type IIa fibers were delayed in recovery at cardiotoxin-induced regeneration models in WT mice

Although the mechanisms of type IIa fiber reduction in BMD mice remain unclear, there is a report about a shift in the muscle fiber type composition in a cardiotoxin (CTX)-induced regeneration model in the TA of WT mice, which suggested the vulnerability of type IIa fibers to CTX injection (*Dalle et al., 2020*). Next, we observed the changes in type IIa, IIb, and IIx fibers in the point on days 0 (CTX (−)), 1, 3, 5, 7, 14, and 28 after CTX injection. HE staining of TA after CTX injection showed extensive muscle fiber necrosis on day 1, infiltration of small mononuclear cells between the necrotic fibers on day 3, appearance of regenerated centronuclear muscle fibers on day 5, and progressive repair of muscle tissues with growth of regenerated muscle fibers on days 7, 14, and 28 after CTX injection (*Figure 5—figure supplement 1*, *left panels*). MYH-1, -2, and -4 immunohistochemistry in the TA after CTX injection revealed elimination of stainability in all type IIa, IIb, and IIx fibers according to muscle necrosis at days 0 and 3 after CTX injection. On days 5 and 7 after CTX injection, only type IIb fibers appeared, with the recovery of regenerated muscle fibers. The recovery of type IIa and IIx fibers was delayed compared to that of type IIb fibers: days 14 and 28 after CTX injection in type IIa and IIx fibers, respectively (*Figure 5—figure supplement 1*, middle and right panels).

## Capillary formation contacting type IIa fiber was altered in BMD mice compared with WT mice

It is hypothesized that vascular dysfunction accompanied by sarcolemmal nNOS reduction is involved in the pathomechanisms of muscle impairment in human DMD (*Miike et al., 1987*; *Thomas et al., 1998*). Thus, we examined the capillaries supporting muscle fibers using PECAM-1 immunohistochemistry at the inner and outer parts of the TA in WT, d45–49, and *mdx* mice at 3 months. WT mice showed 'ring-pattern' capillaries contacting type IIa and IIx fibers at the inner part of TA. On the contrary, capillaries contacting type IIa and IIx fibers showed morphological changes to 'dot-pattern' at the inner part of TA in d45–49 and *mdx* mice (*Figure 6A*). Especially in type IIa fibers, capillaries that circumferentially contact muscle fibers were rich in WT mice, and showed remarkable 'ring-pattern' around type IIa fibers in WT mice. In contrast, around type IIx fibers in WT mice, capillaries contacting muscle fibers were poor compared with those of type IIa fibers, with 'incomplete ring-patterns' around type IIx fibers in WT mice. In d45–49 and *mdx* mice, these circumferentially contacting capillaries were reduced around the type IIa and IIx fibers, and changed to a 'dot-pattern'. Capillaries contacting around type IIb and I fibers showed a 'dot-pattern' even in WT mice (*Figure 6B*). According to these morphological capillary changes, the endothelial area in contact with both type IIa and IIx fibers were decreased in d45–49 and *mdx* mice compared with WT mice. The reduction rate around type IIa fibers (60%) was larger than that around type IIx fibers (35%), reflecting the difference between the 'ring-pattern' around type IIa fibers and the 'incomplete ring-pattern' around type IIx fibers in WT mice (*Figure 6C*).

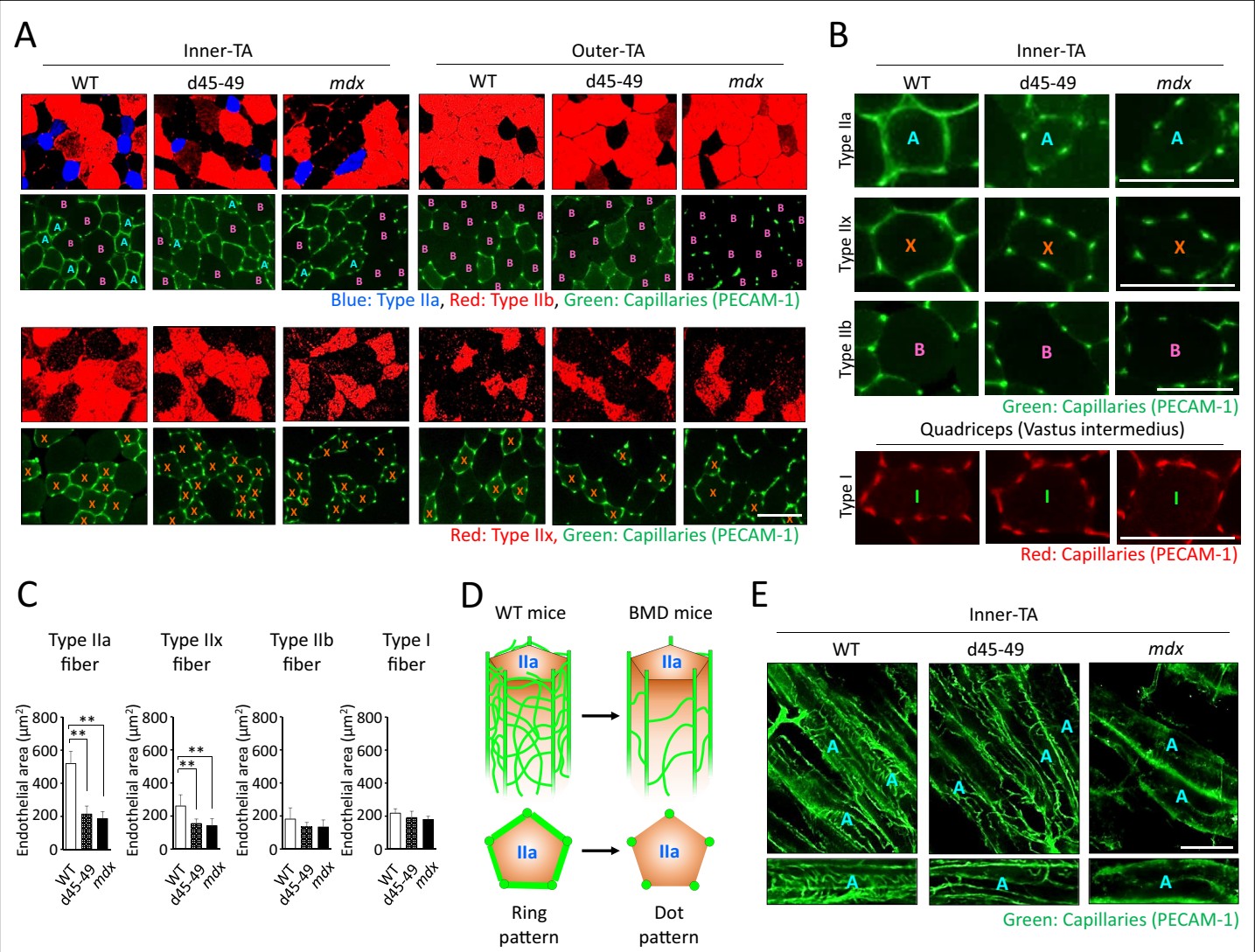

**Figure 6.** Capillaries contacting type IIa fiber were decreased and altered in morphology in Becker muscular dystrophy (BMD) mice. (**A**) MYH-2, -4, -7, and PECAM-1 immunohistochemistry (upper panels: type IIa, IIb fibers and capillaries, respectively) and MYH-1 and PECAM-1 immunohistochemistry (lower panels: type IIx and capillaries, respectively) at the inner and the outer part of tibialis anterior (TA) in wild-type (WT), d45–49, and *mdx* mice, at the age of 3 months. <A>: mean type IIa fibers, <B>: mean type IIb fibers, <X>: mean type IIx fibers. (**B**) High-magnification images of PECAM-1 immunohistochemistry (capillaries) around type IIa, IIx, and IIb fibers at the inner part of TA, and around type I fibers at vastus intermedius in WT, d45–49, and *mdx* mice, at the age of 3 months. <A>: mean type IIa fibers, <B>: mean type IIb fibers, <X>: mean type IIx fibers, <I>: mean type I fibers. (**C**) The Endothelial area (μm²) of capillaries contacting one type IIa, IIx, IIb, and I fiber at the inner part of TA in WT, d45–49, and *mdx* mice, at the ages of 3 months (*n* = 3). (**D**) Schematic representation of an expected mechanisms of morphological changes in capillaries contacting to type IIa fibers from 'ring-pattern' to 'dot-pattern' in BMD mice. (**E**) PECAM-1 immunohistochemistry (capillaries) of the longitudinal sections at the inner part of TA in WT, d45–49, and *mdx* mice, at the age of 3 months (upper panels), and the interconnected branches and capillary loops around a representative type IIa fiber (lower panels). <A>: mean type IIa fibers. Bar: mean ± SD; *p < 0.05, **p < 0.01. Scale bar means 100 μm.

## Discussion

In this study, we established three types of BMD mice carrying the d45–48, d45–47, and d45–49. All BMD mice showed muscle weakness and histopathological changes, including muscle degeneration and fibrosis; however, these changes appeared at different times with each exon deletion. In addition, we confirmed the decreased truncated dystrophin levels and the decreased nNOS expression levels in all BMD mice. Furthermore, unlike *mdx* mice, BMD mice showed site-specific muscle degeneration in particular muscle parts, especially those rich in type IIa fibers.

The phenotypic evaluation confirmed that the d45–49 and *mdx* mice had larger body weights at 1 and 3 months compared with d45–48, d45–47, and WT mice; however, after 6 months, there was no difference in body weight between the groups. *Mdx* mice are generally larger in body weight at a young age than WT mice, whereas the difference between *mdx* and WT mice disappears at old age (*Pastoret and Sebille, 1995*; *Shiba et al., 2015*). The body weights of *mdx* mice in our study were consistent with previous reports, and we demonstrated that the d45–49 mice showed similar changes in body weight as *mdx* mice.

In this study, muscle weakness in forelimb grip strength in *mdx* mice was already apparent from 1 month and was consistent toward 12 months compared with WT mice. In contrast, all BMD mice showed muscle weakness in forelimb grip strength compared to WT mice, but the beginning of the appearance differed depending on the type of exon deletion. Muscle weakness was demonstrated in the d45–49 mice at 1 month similar to that of *mdx* mice, but was apparent in the d45–48 and d45–47 mice after 3 months. All BMD mice reached the same levels of muscle weakness after 6 months whereas milder than those of *mdx* mice. Furthermore, analysis of the hanging wire test revealed decreased power to pull up the trunk in BMD mice, especially in the d45–49 mice. Generally, *mdx* mice show muscle weakness in grip strength (*Takeshita et al., 2017*) and hanging wire tests (*Klein et al., 2012*), and muscle weakness is already seen at 1 month (*McDonald et al., 2015*; *Aartsma-Rus and van Putten, 2014*). Furthermore, *mdx* mice are difficult to pull themselves up from their hanging position (*Hübner et al., 1996*), which was consistent with our results. The BMD rats did not show significant muscle weakness, and the BMD mice carrying the d45–47 showed muscle weakness at 10–15 weeks compared to WT mice, whereas it was milder than that in *mdx* mice (*Teramoto et al., 2020*; *Heier et al., 2023*). On the other hand, we were able to demonstrate differences in severity due to exon deletions in BMD mice, consistent with the severity of human BMD carrying d45–48, d45–47, and d45–49.

The histopathological changes in our *mdx* mice correspond to previous reports (*Pastoret and Sebille, 1995*; *Pessina et al., 2014*; *Giovarelli et al., 2022*). BMD rats showed muscle degeneration accompanied by necrotic fibers and inflammatory cell infiltration at 1 month and became conspicuous after 2 months, and muscle fibrosis was shown after 2 months (*Teramoto et al., 2020*). In addition, BMD mice carrying d45–47 showed muscle degeneration with necrotic fibers and immune cell infiltration, and an increase in fibrotic areas (*Heier et al., 2023*). In this study, all BMD mice exhibited muscle degeneration and fibrosis correspond to previous reports, furthermore, we revealed the timing of its appearance varied depending on the type of exon deletion.

All BMD mice showed faint and patchy dystrophin staining patterns and decreased truncated dystrophin expression levels. These findings are typical in the muscles of patients with BMD (*Beggs et al., 1991*; *Jimi et al., 1992*). Furthermore, sarcolemmal nNOS reduction was observed in all BMD mice to levels similar to those in *mdx* mice, consistent with the lack of an nNOS-binding site from exon deletions. Unlike normal, sarcolemmal nNOS expression is decreased and altered in localization not only in patients with DMD and *mdx* mice (*Brenman et al., 1995*), but also in patients with BMD (*Torelli et al., 2004*; *Chao et al., 1996*). The sarcolemmal expression of aSG was remaining in BMD mice; however, the expression level was slightly decreased. In patients with BMD, proteins expression of DGC including aSG are usually detected in the sarcolemma (*Chao et al., 1996*; *Anthony et al., 2014*). In our BMD mice, utrophin expression was slightly increased compared with WT mice, but this change was smaller than that in *mdx* mice. Utrophin is overexpressed in the muscles of human DMD (*Anthony et al., 2014*; *Arechavala-Gomeza et al., 2010*) and mdx mice (*Pons et al., 1994*; *Banks et al., 2014*), but not in the muscles of patients wih BMD (*Torelli et al., 2004*; *Arechavala-Gomeza et al., 2010*; *Pons et al., 1994*; *Janghra et al., 2016*). All of our BMD mice showed reduced nNOS expression and residual aSG expression on the sarcolemma and no overexpression of utrophin.

Our BMD mice showed differences in the beginning of muscle weakness, muscle degeneration, and fibrosis accompanied by exon deletions, but the mechanisms underlying these differences from exon deletions remain unclear. We confirmed decreased levels of truncated dystrophin in all BMD mice; however, the expression levels were almost the same despite differences in phenotypic severity. Therefore, the severity of BMD in mice might be influenced by the qualitative shift in truncated dystrophin proteins along with exon deletions, and not by quantitative changes. Indeed, using an in silico prediction model, it has been reported that in-frame exon deletions possibly induce a structural shift

in dystrophin, and the collapse of the dystrophin structure might influence BMD severity (*Nicolas et al., 2015*).

We demonstrated site-specific muscle degeneration in BMD mice, unlike *mdx* mice, which, showed diffuse and non-specific muscle degeneration. Even in *mdx* mice, multiple and heterogeneously distributed MRI hyperintensities are seen in a short period of approximately 13–19 weeks (*Pratt et al., 2013*), although BMD mice showed selective and site-specific muscle degeneration, especially in the deep muscle or the inner part of the same muscle. In human BMD, specific muscles are known to change the intensity of muscle MRI images (*Tasca et al., 2012*). Serum CK levels were two- to fourfold higher in all BMD mice than those of WT mice after 3 months and similar to that in a previous study on a BMD rat model (*Teramoto et al., 2020*). However, the difference in serum CK levels between BMD and WT mice was less than that between *mdx* and WT mice. These results may be due to the small area of muscle degeneration in BMD mice because of its uneven distribution in the deep part of each muscle compared with that in *mdx* mice, which showed diffuse muscle degeneration. It has been known that muscles located near the body surface are rich in white muscle fibers, while the deep muscles are rich in medium and red muscle fibers (*Ogata, 1958*). In our BMD mice, type IIa fibers were decreased compared to WT mice, especially in the inner muscle region, where type IIa fibers are abundant. And we found that type IIa fiber reduction started after 3 months in BMD mice, whereas WT mice showed type IIa fiber increment after 3 months. In contrast, type I, IIx, and IIb fibers were unchanged in BMD mice compared with WT mice (*Figure 5C*).

Type IIa fibers are also known as fast-oxidative fibers (*Talbot and Maves, 2016*) and are known to have fatigue resistance compared with type IIb and IIx fibers, known as having the least and the second least fatigue-resistant fibers (*Larsson et al., 1991*) type IIa fiber reduction is thought to contribute to muscle fatigability (*Percival et al., 2010*). Recently, it was reported that mice lacking the RNA-binding protein Musashi-2, which is predominantly expressed in slow-type muscle fibers, showed a reduction in type IIa fibers along with reduced muscle contraction force (*Furuichi et al., 2023*). These findings suggest that decrease in type IIa fibers contributes to the muscle weakness and fatigability in BMD mice. In addition, only type I, IIa, and IIx fibers are present in human skeletal muscle, and type IIb fibers appear to be absent (*Talbot and Maves, 2016*; *Schiaffino and Reggiani, 2011*). Thus, the reduction of type IIa fibers might have a greater effect on the skeletal muscle function of human BMD than that of BMD mice. We examined the association between type IIa fiber reduction and muscle atrophy, but there was no remarkable CSA reduction or changes in muscle atrophy inducing factors. The vulnerability of type IIa fibers to CTX-induced muscle damage has been reported (*Dalle et al., 2020*), and we also observed that the recovery of type IIa and IIx fibers was delayed compared to that of type IIb fibers after CTX injection. This vulnerability and delayed recovery of type IIa fibers may partly explain the type IIa fiber reduction in BMD mice, but the recovery of type IIx fibers was slower than that of type IIa fibers after CTX injection. Therefore, the type IIa fiber-specific decay in BMD mice might not be explained by this vulnerability and delayed recovery during muscle degeneration and regeneration. A decrease in type IIa fibers was also observed in microgravity-induced muscle atrophy in WT mice, where a decrease in sarcolemmal nNOS occurred (*Sandonà et al., 2012*). It has been reported that the expression level of nNOS is higher in type IIa fibers than in type I and IIb fibers (*Planitzer et al., 2001*) and that nNOS-deficient mice have reduced type IIa fibers compared to WT mice (*Percival et al., 2010*) and reduced capillary density only in the inner part of the muscle (*Baum et al., 2023*).

By examining BMD mice, we found for the first time morphological capillary changes from a 'ring-pattern' to a 'dot-pattern' with decreased nNOS expression in the sarcolemma and fewer capillaries in circumferential contact with type IIa fibers. Capillary changes to a 'dot-pattern' were also seen around type IIx fibers in BMD mice, but in WT mice, capillaries around type IIx fibers were poor compared with that of type IIa fibers, and showed an 'incomplete ring-pattern'. In addition, capillaries around type IIb and I fibers showed a 'dot-pattern' even in WT mice. These results suggest that type IIa fibers may require numerous capillaries and maintained blood flow, compared with other muscle fibers, and this high requirement for blood flow might be associated with the type IIa fiber-specific decay in BMD mice. Between the main muscle capillaries running parallel to the muscle fibers, there are transversely interconnected branches and capillary loops (*Haas and Nwadozi, 2015*), and the capillary changes in BMD mice may be associated with the deterioration of these interconnected branches and capillary loops (*Figure 6D*). We examined transversely interconnected branches and capillary loops, using longitudinal muscle sections. We confirmed that there were fewer interconnected capillaries in

BMD and *mdx* mice than in WT mice (*Figure 6E*). Vascular dysfunction has been implicated in muscle damage in canine model of DMD (*Kodippili et al., 2021*) and in human DMD (*Thomas et al., 1998*), and our analysis of BMD mice also suggests that the reduction of type IIa fibers may be influenced by vascular dysfunction with reduced sarcolemmal nNOS as well as fragility and delayed recovery.

## Materials and methods

### Animals

WT control mice (strain: C57BL/6J) were purchased from Jackson Laboratory (Bar Harbor, ME, USA), and dystrophin-deficient *mdx* mice (strain: C57BL/6J) were a gift from the National Center of Neurology and Psychiatry (Tokyo, Japan). The mice were housed in plastic cages in a temperature-controlled environment (23 ± 2°C) with a 12-hr light/dark cycle and free access to food and water. All animal experiments were performed per the institutional guidelines and approved by the Institutional Review Board of Shinshu University, Japan.

### Generation of CRISPR/Cas9-induced BMD mice

To introduce the mutations d45–48, d45–47, and d45–49 in the mouse *Dmd*, we designed four gRNA corresponding to intron sequences 44, 47, 48, and 49 (gRNA-44, -47, -48, and -49, respectively). Next, we electroporated two gRNA at the combination of gRNA-44 and -47 to generate d45–47, gRNA-44 and -48 to generate d45–48, and gRNA-44 and -49 to generate d45–49, with Cas9-nuclease (Integrated DNA Technologies), into embryos from C57BL/6J female mice. F0 mice with the three desired mutations were selected based on the results of PCR analysis and DNA sequencing of tail DNA using intron primers set forward and backward at the deletions. F0 female mice were repeatedly backcrossed with WT male mice, and then F4–F8 male BMD mice having d45–48, d45–47, and d45–49 were used in our study, with WT control and *mdx* mice (male mice; *n* = 3; at the ages of 1, 3, 6, and 12 months). Furthermore, we examined multiple mouse lines with the same exon deletions of d45–48 and d45–47, and confirmed muscle weakness and pathological changes similar to those observed in the mouse line used in this study (data not shown).

### Serum CK

Whole blood was collected from the abdominal aorta under anesthesia with isoflurane in WT, d45–48, d45–47, d45–49, and *mdx* mice at 1, 3, 6, and 12 months (*n* = 3), and were centrifuged at 3000 × *g* for 12 min at 4°C. CK activity in the separated serum was assayed using an automated biochemical analyzer (JCA-BM6050, JEOL) at Oriental Yeast Co, Ltd (Tokyo, Japan).

### Skeletal muscle function

The forelimb grip strength was performed in all mice at 1, 3, 6, and 12 months (*n* = 10 at 1 and 3 months, *n* = 4 at 6 and 12 months) complying with 'Assessing functional performance in the *mdx* mouse model' (*Hübner et al., 1996*). Briefly, mice were placed with their forelimbs on a T-shaped bar and gently pulled backward until their grasps broke. Peak force was automatically recorded using a grip meter (MK-380V, Muromachi). Fifteen tests were performed with a short resting period between each test, and grip strength normalized to body weight was determined by taking the average of the three highest of the 15 values. The hanging wire test was performed at 3 months complying with the same assessments (*Hübner et al., 1996*).

### Muscle tissue extraction and preparation

TA muscles were dissected from WT, d45–48, d45–47, d45–49, and *mdx* mice at 1, 3, 6, and 12 months (*n* = 3), and were frozen in isopentane cooled by liquid nitrogen for histological and immunohisto-chemical analyses and protein and RNA isolation, and were stored at −80°C. Seven-μm-thick transverse cryostat sections were cut in the center of TA muscles, placed on slides, air-dried, and stained with hematoxylin and eosin (HE) and Sirius Red. The quadriceps, erector spinae, diaphragm, and heart muscles were dissected, frozen, and stained with HE as noted above. Sections were viewed and photographed using a BZ-X710 digital camera system (Keyence).

## Immunohistochemical analysis

For immunofluorescence staining, serial 7-μm-thick cross sections of frozen skeletal muscle tissues were mounted on glass slides. The sections were air-dried and blocked in 20% goat serum in phosphate-buffered saline (PBS) for 15 min and incubated with primary antibodies in 5% goat serum in PBS at 4°C overnight. The sections were washed briefly with PBS before incubation with secondary antibodies for 3 hr at 4°C and then washed four times with PBS. The slides were mounted using VECTASHIELD mounting medium (Vector Laboratories) and images were captured using a BZ-X710 digital camera system (Keyence). The primary antibodies were as follows: mouse anti-dystrophin (NCL-DYS2, Leica Biosystems), rabbit anti-nNOS (61-7000, Invitrogen), mouse anti-alpha-sarcoglycan (NCL-a-SARC, Leica Biosystems), mouse anti-MYH-1 (6H1, DSHB), mouse anti-MYH-2 (SC-71, DSHB), mouse anti-MYH-4 (BF-F3, DSHB), mouse anti-MYH-7 (BA-D5, Millipore), and rat anti-PECAM-1 (550274, BD Bioscience).

## Morphometric analysis

Morphometric analysis was performed to identify opaque, necrotic, and centronuclear fibers using HE-stained TA at 1 and 3 months ($n$ = 3), and to identify fiber type composition using MYH-1, -2, -4, and -7 (type IIx, IIa, IIb, and I fibers, respectively) immunohistochemically staind TA, quadriceps, and erector spinae muscles at 3 months ($n$ = 3). At least 500 fibers were analyzed for each muscle, and we calculated the percentages of opaque, necrotic, and centronuclear fibers, and the percentages of type IIx, IIa, IIb, and I fibers. All images were obtained under identical conditions at the same magnification. A study for muscle fibrosis was performed on 7-μm-thick TA muscle sections stained with Sirius Red stain at 1, 3, 6, and 12 months. The area occupied by fibrosis was detected using ImageJ, and the percent fibrosis area was calculated. In addition, studies on capillaries contacting type IIa fibers were performed on MYH-2 and PECAM-1 immunohistochemically stained muscle sections, and the number and size of PECAM-1-positive capillaries contacting type IIa fibers were counted. The endothelial area was calculated using the number and size of capillaries according to previous methods (*Baum et al., 2023*; *Miyazaki et al., 2011*).

## Total protein extract and western blotting

Muscle tissues (20 mg) were homogenized in 150 μl of RIPA buffer (WAKO) containing Halt phosphatase and proteinase inhibitor cocktail (Thermo Fisher Scientific) using an ultrasonic homogenizer (VCX-130; Sonic and Materials INC). After centrifugation (20 min at 15,000 × $g$), the protein concentration in the supernatant was estimated using a BCA Protein Assay Kit (Thermo Fisher Scientific). Protein extracts from each sample were denatured for 5 min at 95°C in NuPAGE LDS Sample Buffer (Thermo Fisher Scientific), and 10 μg/lane protein extracts were submitted to 3–8% NuPAGE Novex Tris-acetate gel electrophoresis (Thermo Fisher Scientific) with HiMark pre-stained standard proteins (Thermo Fisher Scientific) at 150 V for 70 min. The resulting gel was transferred onto a 0.2-μm nitrocellulose membrane (Bio-Rad) at 400 mA for 40 min using an EzFastBlot HMW buffer (ATTO). Membranes were incubated with primary antibodies and peroxidase-conjugated secondary antibodies (Bio-Rad) using an iBind Flex Western Device (Thermo Fisher Scientific). All membranes were visualized using ECL Prime western blotting detection reagent (Cytive) and ChemiDoc (Bio-Rad). Band intensity and total protein normalization were determined using Image Lab software (Bio-Rad). The primary antibodies were as follows: rabbit anti-dystrophin (ab15277, Abcam) and mouse anti-utrophin (sc-33700, Santa Cruz Biotechnology).

## RNA isolation and RT-PCR

Frozen muscle tissues (20 mg) were homogenized and total RNA was isolated using the RNeasy Fibrous Tissue Kit (QIAGEN). cDNA was synthesized using the QuantiTect Reverse Transcription Kit (QIAGEN, Hilden, Germany). The levels of mRNA and 18 S rRNA were quantified by qPCR using Fast SYBR Green Master Mix (Thermo Fisher Scientific) and QuantStudio 3 (Thermo Fisher Scientific) with 10 nM of each primer at a final volume of 10 μl. Thermal cycling conditions for all primers were 10 min at 95°C, then 40 cycles each of 15 s at 94°C and 30 s at 60°C. Each mRNA quantity was calculated using the delta-delta-CT method with 18S rRNA as the housekeeping gene. The primer sequences used for RT-PCR are shown in *Supplementary file 1*.

## CTX muscle injury and histochemistry

We injected 100 µl of CTX (10 mM in 0.9% NaCl) (Sigma-Aldrich) into the TA muscle of WT mice at 6 weeks of age using a 27-gauge needle and a 1-ml syringe under anesthesia with isoflurane according to the previous methods (*Miyazaki et al., 2011*). The TA muscles were isolated before the injection, and 1, 3, 7, and 14 days after (*n* = 1); 7 µm transverse cryostat sections of frozen muscles were stained with HE and with MYH-1, -2, -4, and -7 immunohistochemistry.

## Statistical analysis

Results are expressed as ± SEM. Statistical analysis was performed to establish the significance between groups using a one-way ANOVA. Intergroup comparisons were performed using Bonferroni correction. Statistical significance was set at $p < 0.05$. Statistical analyses were performed using GraphPad Prism version 9.0 (GraphPad Software Inc, La Jolla, CA, USA).

## Acknowledgements

We thank K Kimura (University of Tokyo, Tokyo, Japan) and A Hineno (Shinshu University School of Medicine, Matsumoto, Japan) for the helpful discussions and advice. We thank S Takeda (National Center of Neurology and Psychiatry, Tokyo, Japan) for providing *mdx* mice. Financial support for this work was provided by Grants-in-Aid for Scientific Research (C) from the Ministry of Education, Culture, Sports, Science, and Technology of Japan (No. 22K07021).

## Additional information

### Funding

| Funder | Grant reference number | Author |
| --- | --- | --- |
| Ministry of Education, Culture, Sports, Science, and Technology of Japan | No. 22K07021 | Daigo Miyazaki |

The funders had no role in study design, data collection and interpretation, or the decision to submit the work for publication.

### Author contributions

Daigo Miyazaki, Conceptualization, Data curation, Formal analysis, Funding acquisition, Writing – original draft; Mitsuto Sato, Data curation, Visualization; Naoko Shiba, Data curation, Methodology; Takahiro Yoshizawa, Data curation, Formal analysis, Methodology; Akinori Nakamura, Conceptualization, Data curation, Project administration, Writing - review and editing

### Author ORCIDs

Daigo Miyazaki ⓘ https://orcid.org/0000-0002-0032-0077
Akinori Nakamura ⓘ http://orcid.org/0000-0001-8801-6688

### Ethics

All animal experiments were performed per the institutional guidelines and approved by the Institutional Review Board of Shinshu University, Japan (No. 023073).

Reviewer #2 (Public review): https://doi.org/10.7554/eLife.100665.3.sa1
Author response https://doi.org/10.7554/eLife.100665.3.sa2

## Additional files

### Supplementary files

Supplementary file 1. The list of primer sequences used for RT-PCR in this study.

## Data availability

All data generated or analyzed during this study are included in the manuscript and supporting files; data files for primers used in RT-PCR have been provided as supplementary file 1.

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
