## [Editor Report · eLife Assessment]

The authors present three transgenic models carrying three representative exon deletions of the dystrophin gene. The findings presented are **valuable** to the field of muscle diseases, particularly muscular dystrophies. The evidence provided in the manuscript is **convincing**, with rigorous biochemical assays and state-of-the-art microscopy methods.

---

## [Referee Report · Reviewer #2 (Public review)]

Miyazaki et al. established three distinct BMD mouse models by deleting different exon regions of the dystrophin gene, observed in human BMD. The authors demonstrated that these models exhibit pathophysiological changes, including variations in body weight, muscle force, muscle degeneration, and levels of fibrosis, alongside underlying molecular alterations such as changes in dystrophin and nNOS levels. Notably, these molecular and pathological changes progress at different rates depending on the specific exon deletions in dystrophin gene. Additionally, the authors conducted extensive fiber typing, revealing a site-specific decline in type IIa fibers in BMD mice, which they suggest may be due to muscle degeneration and reduced capillary formation around these fibers.

Strengths:

The manuscript introduces three novel BMD mouse models with different dystrophin exon deletions, each demonstrating varying rates of disease progression similar to the human BMD phenotype. The authors also conducted extensive fiber typing across different muscles and regions within the muscles, effectively highlighting a site-specific decline in type IIa muscle fibers in BMD mice.

Comments on revisions:

The authors did an excellent job addressing all or most of the concerns I raised in my previous review and have incorporated the necessary changes into the manuscript.

---

## [Author Response]

The following is the authors’ response to the original reviews.

**Public Reviews:**

**Reviewer #1 (Public review):**
Summary:In this article the authors described mouse models presenting with backer muscular dystrophy, they created three transgenic models carrying three representative exon deletions: ex45-48 del., ex45-47 19 del., and ex45-49 del. This article is well written but needs improvement in some points.Strengths:This article is well written. The evidence supporting the authors' claims is robust, though further implementation is necessary. The experiments conducted align with the current state-of-the-art methodologies.Weaknesses:This article does not analyze atrophy in the various mouse models. Implementing this point would improve the impact of the work

We thank the reviewer for their constructive suggestions and comments on this work. Muscle hypertrophy is shown with growth in dystrophin-deficient skeletal muscle in mdx mice; thus, we did not pay attention to the factors associated with muscle atrophy in BMD mice. As the reviewer suggested, the examination of the association between type IIa fiber reduction and muscle atrophy is important, and the result is considered to be helpful in resolving the cause of type IIa fiber reduction in BMD mice.

In response, we reviewed the following.

(1) The cross-sectional areas (CSAs) of muscles. We confirmed that the CSAs in BMD and *mdx* mice were rather high at 3 months, in accordance with muscle hypertrophy, compared with those of WT mice. The data is presented in Fig. 4–figure supplement 1B.

(2) The mRNA expression levels of Murf1 and atrogin-1. We confirmed that these muscle atrophy inducing factors did not differ among WT, BMD, and *mdx* mice. The data is presented in Fig. 4–figure supplements 1C and 1D.

**Reviewer #2 (Public review):**
Summary:Miyazaki et al. established three distinct BMD mouse models by deleting different exon regions of the dystrophin gene, observed in human BMD. The authors demonstrated that these models exhibit pathophysiological changes, including variations in body weight, muscle force, muscle degeneration, and levels of fibrosis, alongside underlying molecular alterations such as changes in dystrophin and nNOS levels. Notably, these molecular and pathological changes progress at different rates depending on the specific exon deletions in the dystrophin gene. Additionally, the authors conducted extensive fiber typing, revealing a site-specific decline in type IIa fibers in BMD mice, which they suggest may be due to muscle degeneration and reduced capillary formation around these fibers.Strengths:The manuscript introduces three novel BMD mouse models with different dystrophin exon deletions, each demonstrating varying rates of disease progression similar to the human BMD phenotype. The authors also conducted extensive fiber typing across different muscles and regions within the muscles, effectively highlighting a site-specific decline in type IIa muscle fibers in BMD mice.Weaknesses:The authors have inadequate experiments to support their hypothesis that the decay of type IIa muscle fibers is likely due to muscle degeneration and reduced capillary formation. Further investigation into capillary density and histopathological changes across different muscle fibers is needed, which could clarify the mechanisms behind these observations.

We thank the reviewer for these positive comments and the very important suggestion about type IIa fiber reduction and capillary change around muscle fibers in BMD mice. From the results of the cardiotoxin-induced muscle degeneration and regeneration model, type IIa and IIx fibers showed delayed recovery compared with that of type-IIb fibers. However, this delayed recovery of type IIa and IIx could not explain the cause of the selective muscle fiber reduction limited to type IIa fibers in BMD mice. Therefore, we considered vascular dysfunction as the reason for the selective type IIa fiber reduction, and we found morphological capillary changes from a “ring pattern” to a “dot pattern” around type IIa fibers in BMD mice. However, the association between selective type IIa fiber reduction and the capillary change around muscle fibers in BMD mice remains unclear due to the lack of information about capillaries around type IIx and IIb fibers. The reviewer pointed out this insufficient evaluation of capillaries around other muscle fibers (except for type IIa fibers), and this suggestion is very helpful for explaining the association between selective type IIa fiber reduction and vascular dysfunction in BMD mice.

In response, we reviewed the following.

(1) The capillary formation around type IIx, IIb, and I fibers, in addition to that around type IIa fibers. We found that capillaries contacting around type IIx, IIb, and I fibers were poor in WT mice compared with that around type IIa fibers, with ‘incomplete ring-patterns’ around type IIx fibers, and ‘dot-patterns’ around type IIb and I fibers in WT mice. Morphological capillary changes around muscle fibers from WT to d45-49 and *mdx* mice were ‘incomplete dot-pattern’ to ‘dot-pattern’ around type IIx fibers, and ‘dot-pattern’ to ‘dot-pattern’ around type IIb and I fibers. This was in contrast to those around type IIa fibers: remarkable ‘ring-pattern’ to ‘dot-pattern’. These data are presented in Fig. 6B.

(2) The endothelial area in contact with type IIx, IIb, and I fibers, and additionally that in contact with type IIa fibers. The endothelial area in contact with both type IIa and IIx fibers was less in d45-49 and *mdx* mice than in WT mice, but the reduction was larger around type IIa fibers than around type IIx fibers, reflecting the difference between the ‘ring-pattern’ around the former and the ‘incomplete ring-pattern’ around the latter in WT mice. These data are presented in Fig. 6C.

(3) Transversely interconnected branches and capillary loops, using longitudinal muscle sections. We confirmed that there were fewer interconnected capillaries in BMD and *mdx* mice than in WT mice. These data are presented in Fig. 6E.

(4) The mRNA expression levels of neuronal nitric oxide synthase (nNOS). We confirmed that nNOS protein expression levels were decreased in BMD and *mdx* mice in spite of adequate levels of nNOS mRNA expression. The data on nNOS mRNA expression levels is presented in Fig. 3–figure supplement 1C.

(5) We added a sentence in the Abstract about the potential utility of BMD mice in developing vascular targeted therapies.

**Recommendation for the authors:**

**Reviewer #1 (Recommendation for the authors):**
Abstract:Abstract: more emphasis should be on the pathological implications of Becker muscular dystrophy (BMD). Furthermore, should be emphasized the findings made in this article and the conclusions. Abbreviations such as DMD and MDX should be written in full and only then with the acronym.

We appreciate the reviewers’ comments, and we apologize for the confusion over abbreviations. *DMD* is the gene name encoding dystrophin, and *mdx* is the strain name of mouse lacking dystrophin.

In the Abstract and the Figure legends we changed:

(1) DMD to *DMD*;

(2) mdx mice to *mdx* mice.

Results:Line 95: in this line, authors evaluated serum creatinine kinase (CK) levels at 1, 3, 6 and 12 months in WT mice and mdx mice. Why did you decide to study it? This part should be described in more detail. Serum CK is one of the main markers of muscle necrosis; therefore, I would report this data alongside the description of the muscle histology and necrotic fibers.

We thank the reviewers for the important remarks. In this study, serum creatine kinase (CK) levels were two-fold to four-fold higher in BMD mice than in WT mice, but its rate of increase was less than that of *mdx* mice. We consider that the lesser changes in serum CK levels in BMD mice may be due to the smaller area of muscle degeneration because of focal and uneven muscle degeneration compared with that in *mdx* mice, which showed diffuse muscle degeneration.

In response, we have moved the description of serum CK levels in the Results, from the section about the establishment of BMD mice to the section about site-specific muscle degeneration in BMD mice.

In addition, we added a description in the Discussion about the possible association between the lesser changes in serum CK levels in BMD mice and its uneven distribution of muscle degeneration.

Line 192-202: In these lines, authors observed a decrease in type IIa fibers after 3 months in BMD mice. I suggest evaluating also atrophy through evaluating cross-sectional areas (CSA) and expression of Murf1 and Atrogin1

We thank the reviewer for the point about the association between type IIa fiber reduction and muscle atrophy. We evaluated the CSAs and the mRNA expression levels of Murf1 and atrogin-1. We confirmed that the CSAs in BMD and *mdx* mice were rather high at 3 months, in accordance with muscle hypertrophy, compared with those of WT mice, and that Murf1 and atrogin-1 mRNA expression levels did not differ among WT, BMD, and *mdx* mice. These data are presented in Fig. 4–figure supplements 1B, 1C, and 1D. We added a sentence about the changes in CSA and muscle atrophy inducing factors in the Discussion.

Methods and materialLine 342-348: authors have described animals, but not specified sex and number of mice in each group. This part should be improved.

We apologize for our insufficient information about the sex and number of mice in the Materials and methods.

We added a sentence specifying the sex, number, and evaluation period of each mouse group in the section on the generation of BMD mice.

Line 426-433: authors described qPCR. It is necessary that the authors also describe primer sequences.

We apologize for any lack of information about the primer sequences used in qPCR analysis. Supplemental Table 1 lists the primer sequences.

We also added a sentence about the information in the primer list in the section on RNA isolation and RT-PCR in the Materials and methods.

**Reviewer #2 (Recommendation for the authors):**
Miyazaki et al. established three distinct BMD mouse models by removing different exon regions of the dystrophin gene. The authors demonstrated that the pathophysiological and molecular changes in these models progress at varying rates. Additionally, they observed a site-specific decline in type IIa fibers in BMD mice, while the proportions of other fiber types, such as type I and type IIx, remained consistent with those in wild-type mice. They proposed that the selective decay of type IIa fibers in BMD mice could be due to two primary factors: (1) muscle degeneration and regeneration, supported by their findings in cardiotoxin-treated mouse models, and (2) reduced capillary formation around type IIa fibers. However, the authors also presented evidence that type IIx fibers exhibited delayed recovery, similar to type IIa fibers, as demonstrated in cardiotoxin-induced regeneration models. Additionally, dot-patterned capillary formations were observed around both type IIa and type IIx fibers. Despite these findings, BMD mice did not show any changes in the proportion of type IIx fibers in inner BMD muscles. The authors should consider adding further analysis to strengthen their hypothesis and to disclose any possible mechanisms that led to these discrepancies.If the authors hypothesize that reduced capillary density around type IIa fibers contribute to their site-specific decay in BMD mice, they should consider measuring and statistically analyzing the endothelial area around all fiber types. By plotting and comparing these measurements across different fiber types between wild-type, BMD, and mdx mice, the authors could provide more robust evidence to support their hypothesis. This approach would help clarify whether reduced capillary density is a contributing factor to the site-specific decay of type IIa fibers in BMD mice and the more diffuse, non-specific muscle changes observed in mdx mice.The authors reported in the first part of the manuscript that histopathological changes, including muscle degeneration in BMD mice, are predominantly restricted to the inner part of the muscles. In the second part, they noted a decline in type IIa fibers specifically in the inner muscle region. To strengthen the hypothesis that the decay of type IIa fibers in the inner muscle is linked to muscle degeneration, the authors should consider performing histopathological measurements across different fiber types within the inner muscle. Reporting the correlations between these measurements would provide more compelling evidence to support their hypothesis.

We thank the reviewer for these important suggestions about the association between type IIa fiber reduction and capillary change around muscle fibers in BMD mice. We prepared an additional evaluation about the capillary formation (in Fig. 6B) and endothelial area (in Fig. 6C) around type IIx, IIb, and I fibers. We found that capillaries contacting around type IIx, IIb, and I fibers were poor in WT mice compared with those around type IIa fibers, and showed an ‘incomplete ring-pattern’ around type IIx fibers and a ‘dot-pattern’ around type IIb and I fibers in WT mice, in contrast with type IIa fibers, which showed remarkable ‘ring-pattern’ capillaries. Reflecting this, the changes in endothelial area around type IIx, IIb, and I fibers between WT and BMD mice were less than those around type IIa fibers. These results suggest that type IIa fibers may require numerous capillaries and maintained blood flow compared with type IIx, IIb, and I fibers, and this high requirement for blood flow might be associated with the type IIa fiber-specific decay in BMD mice.

We added the following.

(1) Sentences in the Results about the capillary changes around type IIx, IIb, and I fibers in WT, d45-49, and *mdx* mice.

(2) Sentences in the Results about the changes in endothelial area around type IIx, IIb, and I fibers in WT, d45-49, and *mdx* mice.

(3) Sentences in the Discussion about the association between the type IIa fiber-specific decay in BMD mice and the differences in capillary changes of each muscle fiber from WT to BMD mice.

We changed a sentence in the Discussion about the delayed recovery of type IIa and IIx fibers after CTX injection, to make it clear that the recovery of type IIx fibers was slower than that of type IIa fibers after CTX injection, and that therefore the type IIa fiber-specific decay in BMD mice might not be explained by this vulnerability and delayed recovery during muscle degeneration and regeneration.

Minor Issues:Line 103: The word "mice" is duplicated and should be corrected.

We apologize that “mice” was duplicated. We have corrected it.

Line 120: Revise for clarity: "The proportion of opaque fibers is significantly different between d45-48 mice and WT at 3 months, with an increased tendency observed only in 1-month-old mice."

We apologize for the confusion about the proportion of opaque fibers. We revised this sentence as follows.

“Opaque fibers, which are thought to be precursors of necrotic fibers, increased at an earlier age of 1 month in d45–49 mice compared with WT mice; in contrast, the proportion of opaque fibers differs significantly between d45–47 and WT mice at 3 months, with an increased tendency only in 1-month-old mice (Fig. 2C).”

Line 152: Clarify the statement regarding utrophin levels, as it currently contradicts the Western blot data. The sentence reads: "The increased levels of utrophin are 8-fold higher at 1 month and 30-fold higher at 3 months." This should be verified against the data, as the band densities in the Western blots suggest otherwise.

We apologize for the confusion about utrophin expression levels. We revised this sentence as follows.

“By western blot analysis, the utrophin expression levels showed only an increased tendency in all BMD mice at 3 months, whereas there was a significant increase in *mdx* mice (8-fold at 1 month, and 30-fold at 3 months) compared to WT mice (Figs. 3C and F).”

Line 235: Correct the sentence to accurately reflect the findings: "BMD mice showed reduced muscle weakness."

We apologize for our incorrect wording. We have removed the word “reduced” in this sentence.